# Clinical and Mechanistic Implications of R-Loops in Human Leukemias

**DOI:** 10.3390/ijms24065966

**Published:** 2023-03-22

**Authors:** Seo-Yun Lee, Kyle M. Miller, Jae-Jin Kim

**Affiliations:** 1Department of Life Science and Multidisciplinary, Genome Institute, Hallym University, Chuncheon 24252, Republic of Korea; seoyun@hallym.ac.kr; 2Department of Molecular Biosciences, The University of Texas at Austin, Austin, TX 78712, USA; kyle.miller@austin.utexas.edu

**Keywords:** R-loop, cancer, leukemia, transcription, genome instability

## Abstract

Genetic mutations or environmental agents are major contributors to leukemia and are associated with genomic instability. R-loops are three-stranded nucleic acid structures consisting of an RNA–DNA hybrid and a non-template single-stranded DNA. These structures regulate various cellular processes, including transcription, replication, and DSB repair. However, unregulated R-loop formation can cause DNA damage and genomic instability, which are potential drivers of cancer including leukemia. In this review, we discuss the current understanding of aberrant R-loop formation and how it influences genomic instability and leukemia development. We also consider the possibility of R-loops as therapeutic targets for cancer treatment.

## 1. Introduction

Leukemia is a blood cancer or hematologic malignancy that begins in the bone marrow when blood cells proliferate abnormally. There are four types of leukemia: acute lymphoblastic leukemia (ALL), acute myeloid leukemia (AML), chronic lymphocytic leukemia (CLL), and chronic myeloid leukemia (CML). The specific cause of leukemia is unclear; however, genetic mutations and environmental agents, including smoking, ionizing radiation, viruses, and chemicals, are believed to contribute to their development [1,2,3,4]. These insults directly or indirectly cause DNA breaks and it has been estimated that each human cell is subjected to approximately 70,000 DNA lesions per day [5,6]. Although most of these are single-strand DNA (ssDNA) breaks, DNA double-strand breaks (DSBs) do occur at around one break per cell per hour and are potentially more dangerous to the cell. DSBs result from exogenous or endogenous DNA-damaging agents, including X-rays, ultraviolet (UV) light, toxic chemicals, and DNA transcription and/or replication stress. Although both exogenous and endogenous DNA damage have the potential to promote cancer through DNA mutations, the relative contribution of intrinsic and extrinsic factors to cancer incidence remains poorly defined [7,8].

Recent genetic studies have identified intrinsic factors that also contribute to somatic mutations in various cancers [9,10]. R-loop formation is a major source of transcriptional stress [11,12]. R-loops can form during transcription, replication, and DNA repair processes, where they participate in several physiological processes including gene expression regulation through several mechanisms. Genome-wide mapping studies have indicated that R-loops are abundant in promoters [13,14,15] and facilitate transcription by inhibiting DNA methylation [16,17]. DNA methyltransferases bind poorly to RNA–DNA hybrids, which enables R-loops to suppress methylation-associated gene silencing. R-loops also modulate chromatin remodelers in the promoter region by recruiting activating chromatin remodelers, such as histone acetyltransferases, and inhibit repressive chromatin modifying enzymes, including the PRC2 complex, which maintains a poised chromatin state. R-loops also promote the expression of differentiation genes in mouse embryonic stem cells [18].

Although R-loops are essential mediators of transcription, replication, and DNA repair, unregulated R-loop formation is a major source of genomic instability and potential tumorigenesis [19,20,21]. Recent genetic and biochemical studies have identified numerous proteins involved in R-loop regulation, including helicases (e.g., Senataxin), topoisomerases, DNA repair proteins (e.g., BRCA1/2), and RNA splice factors [11]. R-loop-mediated genome instability can also contribute to hematologic malignancies. In this review, we discuss how R-loops affect genome instability and tumorigenesis in hematologic malignancies. We provide an updated and integrated view of the clinical implications of R-loops in leukemia.

## 2. R-Loop-Mediated DNA Damage and Genomic Instability

### 2.1. How Do R-Loops Generate DNA Breaks?

R-loops frequently occur in the genome and contribute to various biological processes, such as immunoglobulin class switching, chromatin organization, mitochondrial DNA replication, and transcriptional regulation [12,22,23]. Although R-loops have been detected across the genome, the generation of excessive and aberrant R-loops can be toxic to the genome and contribute to certain diseases, such as neurodegeneration and cancer [24]. Therefore, it is essential to control R-loop homeostasis and to understand how this may contribute to human diseases. The general source of RNA/DNA hybrids is thought to be created by transcription and DNA replication. During transcription, nascent RNA transiently hybridizes with template DNA within the transcription bubble to produce a transient RNA/DNA hybrid, which is resolved by extruding the newly synthesized transcript from the transcription bubble. Replication-associated RNA/DNA hybrids are caused by lagging strand replication. During the replication process, helicases create a forked DNA structure and DNA polymerase-α (Polα)–primase generates short strands of RNA complementary to single-strand DNA on the lagging strand [25,26]. These RNA/DNA hybrids are subsequently resolved by Okazaki fragment maturation whereby RNA primers are normally removed by flap endonuclease 1 (FEN1) and DNA is linked together by DNA ligase 1 (LIG1) [27].

RNA loops can also be a source of genomic instability when they accumulate in the genome. The displaced non-template ssDNA strand in the R-loop can introduce a DNA mutation and DNA breakage. A group of cytidine deaminases, AID/APOBEC, convert cytosine to uracil in ssDNA and can result in a nick in the DNA (Figure 1a) [28,29]. AID can drive lymphomagenesis by this off-target DNA break generation [30]. These residues are processed by DNA repair factors, which catalyze end-joining and class switching. Nevertheless, inefficient DNA repair during class switching can contribute to genomic instability [23,31,32]. Indeed, ssDNA can be cleaved by DNA nucleases, including XPG, XPF, FEN1, and DNA2, to generate single-strand DNA breaks (SSB) (Figure 1b) [33,34]. G-quadruplex and hairpin secondary DNA structures are also formed by R-loop-mediated ssDNA, which can trigger DNA damage and genome instability (Figure 1c) [35,36,37]. Transcription and replication stress are also well-known sources of R-loops (Figure 1d,e). Although R-loops are essential for transcriptional regulation, they also interfere with transcription when they become uncontrolled. Pausing or stalling the RNA polymerase machinery due to obstacles in the DNA template can trigger R-loop accumulation. DNA lesions also lead to transcription stress and activation of transcription-coupled nucleotide excision repair (TC-NER) [38,39]. During DNA replication, R-loops can occur through replication fork stalling and collision with transcription. The transcription and replication machinery use the same DNA template in S-phase; thus, they may coincide at specific genomic regions. Therefore, transcription-replication conflicts (TRC) are unavoidable during S-phase. The TRC leads to a stalling of the replication forks, which activates the ATR pathway to orchestrate a cellular stress response [40]. R-loop-associated ATR activation acts to prevent excessive fork processing mediated by DNA nucleases (e.g., MUS81) [41]. Thus, because improper R-loop accumulation by diverse processes contributes to DNA damage and genomic instability, it is necessary for the cell to regulate R-loop homeostasis to maintain genome integrity.

### 2.2. R-Loops and Human Cancers

Recent studies have revealed molecular insights into the role of R-loops in human cancers. Many cancers exhibit dynamic oncogene signaling as well as high levels of mutagenesis and genomic instability. Global perturbations in transcription, replication, and RNA processing by oncogenes result in improper R-loop accumulation and the associated DNA damage is linked to mutations that can contribute to the etiology of cancer. Several studies have identified the functional relevance between R-loops and cancer initiation [20,42]. Indeed, a dysregulation of factors related to R-loop resolution results in various types of DNA damage and genomic instability, which can trigger cancer progression [43]. BRCA1/2, which are major regulators of genome maintenance, are mutated in some cancers that show high levels of R-loops and altered oncogene expression, including breast cancer [42,44,45,46]. Moreover, since BRCA1/2 functions along with FA proteins in DNA integrity pathways, their deficiency results in the accumulation of DNA damage and R-loops due to interference between the FA-BRCA pathway. This has been observed to increase the risk of cancer, such as in AML, esophageal, gastrointestinal tract, and head-neck cancer [47,48,49,50,51,52]. Mutations in multiple splicing factors also increase R-loop formation that activate the ATR-mediated DNA damage response in pre-leukemic myelodysplastic syndromes [53]. Emerging evidence supports targeting R-loop as a cancer treatment strategy due to their dysregulation in cancer. Despite the association of R-loops and cancer, the biological processes associated with R-loops in leukemia remain unclear.

In the next section, we describe R-loop regulators and related mechanisms in leukemia specifically (Table 1). New insights revealing how R-loops contribute to leukemogenesis may provide a better understanding of this disease and deliver new strategies for leukemia treatment.

## 3. *HOTTIP*-Dependent R-Loop Formation Induces Leukemogenesis

Long non-coding RNAs (lncRNAs) are important regulators of leukemogenesis. *HOTTIP* (HOXA transcript at the distal tip) is a HOXA locus-associated lncRNA, which regulates transcription of the 5′ end of HOXA genes [62,63]. *HOX* genes represent a subset of homeobox genes that encode for a highly conserved family of homeodomain-containing transcription factors that regulate embryonic development. Their expression levels are temporally and spatially restricted, because the aberrant expression of *HOX* genes results in abnormal development and tumorigenesis [64,65,66,67]. Multiple *HOX* genes, including *HOXA4*, *HOXA5*, *HOXA7*, and *HOXA9*, are upregulated in MLL translocated lymphoblastic leukemias [68,69]. *HOXA9* is required for the survival of MLL-rearranged acute leukemias and its overexpression is a marker of poor prognosis in leukemia patients [70,71,72].

In mixed-lineage leukemia (MLL), rearranged (MLLr+), or nucleophosmin 1 (NPM1)-mutated (NPM1c+) acute myeloid leukemia, *HOTTIP* expression is upregulated, which is associated with a poor survival rate in AML patients [63]. The transgenic overexpression of *HOTTIP* in mice promotes the self-renewal of hematopoietic stem cells (HSC), which leads to an AML-like phenotype by the HOXA topologically associated domain (TAD) and its transcription. Luo et al. demonstrated that *HOTTIP* binds to CTCF-binding sites (CBSs) through the formation of R-loops in the AML genome (Figure 2a). R-loops can be generated co-transcriptionally (in cis) or post-transcriptionally (in trans). If the RNA transcript has homology to other regions, it can form R-loops not only at proximal regions, but also at distally located, sequence-related loci [73,74]. *HOTTIP* is generated at different genomic regions, but it can bind to CBSs to form R-loops [54]. CTCF-cohesion recruitment to WNT pathway targeted oncogenes, including *MYC*, *CTNNB1*, and *MECOM* was shown to be dramatically decreased in *HOTTIP*-depleted cells, whereas global CTCF binding was unaffected. In *HOTTIP*-depleted cells, TAD formations were observed to be decreased in 303 genes involved in WNT signaling, *HOX* gene regulation, AML, and cell-cycle progression. If cells overexpressed dCas9-RNaseH to remove local R-loop formation, oncogene expression decreased [54]. Overall, *HOTTIP*-mediated R-loops appear to recruit CTCF to CBSs, which further reinforces CTCF chromatin boundary activity to promote TAD formation that drives oncogene transcription.

## 4. Depletion of TET Promotes R-Loop Formation-Induced B-Cell Lymphomas

Ten-eleven translocation (TET) enzymes are frequently mutated or functionally inactivated in diffuse large B-cell lymphoma (DLBCL) as well as leukemia [75,76,77,78,79,80]. *TET2* is mutated in ~10% and 24~32% of DLBCL and AML, respectively [81,82,83]. The TET enzymes are methylcytosine dioxygenases, which induce DNA demethylation. They act by catalyzing the stepwise oxidation of DNA 5-methylcytosine (5mc) to 5-hydroxymethylcytosine (5hmc) and further oxidation of 5hmc to 5-formylcytosine (5fc) and then to 5-carboxylcytosine (5caC). 5fc and 5caC can be removed by base excision repair and replaced with cytosine in the base sequence. TET proteins play an important role in transcriptional regulation because demethylation is required for the recruitment of transcription factors to promotor regions to activate transcription. Shukla V et al. examined the effect of depleted TET protein-induced R-loops in B-cell lymphoma (Figure 2b) [55]. In this study, the deletion of the *TET2* and *TET3* genes in mature B cells resulted in germinal center (GC)-derived B-cell lymphomas with increased G-quadruplexes and R-loops. R-loops were found to accumulate and induce DNA damage at immunoglobulin switch regions. To identify the role of DNA demethylation in R-loop accumulation and DNA damage in DLBCL, DNA methyltransferase 1 (DNMT1) was deleted in TET-deficient B cells. The deletion of DNMT1 prevented the accumulation of R-loop and delayed B lymphoma development, which indicates that DNA demethylation is necessary for R-loop suppression and genome integrity during B-cell development. However, it remains unclear how DNA methylation controls R-loop accumulation at immunoglobulin switch regions. Therefore, further studies are needed to fully understand how R-loops are regulated by DNA methylation and demethylation. Shukla et al. analyzed R-loop-mediated DNA damage and tumorigenesis in B-cell lymphoma (non-Hodgkin lymphomas). The frequent occurrence of TET and DNMT1 mutations in leukemia suggests that genetic instability caused by R-loops could contribute to leukemia development, including through alterations in DNA methylation pathways [81,84]. Similar studies are needed in leukemic cells to identify the roles of R-loops in tumorigenesis in this cancer context.

## 5. R-Loops Regulate Transcription through DNA Demethylation

As described above, DNA methylation promotes R-loop formation; however, R-loops also suppress DNA methylation at CpG islands to regulate transcription. Senataxin unwinds RNA/DNA hybrids through its helicase activity [85,86,87]. Senataxin mutations are found in motor neuron disease (amyotrophic lateral sclerosis 4, ALS4) [88]. The overexpression of a gain-of-helicase functional mutant in Senataxin resulted in decreased R-loop formation, which was associated with repression of gene expression by DNMT1 [17]. DNMT1 has a significantly lower affinity for RNA/DNA hybrids compared with double-stranded DNA, which indicates that R-loop formation suppresses DNMT binding to CpG islands. Global R-loop mapping in the human genome revealed that R-loop formation is strongly correlated with nonmethylated CpG islands and DNMT3B methylation is blocked when transcribed through an R-loop-forming sequence [16]. These results suggest that R-loops prevent DNA methylation at CpG islands to promote transcription. The methylation of promotor regions can promote tumorigenesis by suppressing tumor suppressor gene expression [89,90]. DNMT inhibitors have been developed and evaluated in clinical trials [91,92] for AML to restrain unregulated DNMT-mediated DNA methylation. DNMT3A is also frequently mutated in this cancer [84]. The disruption of normal R-loop levels and mutation of DNMT3A may affect DNA methylation in leukemia cells. This result may indicate that normal R-loop levels are essential regulators for the R-loop–DNMT–DNA methylation–tumorigenesis axis. Whether or not R-loops impact the known functions of DNA methylation in genome integrity pathways in addition to transcription awaits further investigation [93].

## 6. Human T-Cell Leukemia Virus Type 1 (HTLV-1) Promotes R-Loop Formation and Adult T-Cell Lymphomas

Human T-cell leukemia virus type 1 (HTLV-1) is a retrovirus of the human T-lymphotropic virus (HTLV) family, which induces very aggressive adult T-cell leukemia/lymphoma (ATL) [94,95,96]. HTLV-1 is also associated with myelopathy, uveitis, Strongyloides stercoralis hyperinfection, and other diseases. It does not contain a cellular oncogene, but it encodes Tax, which functions as a transactivator protein. Tax induces constitutive NF-κB activation through a protein complex with IκB kinase (Taxisome) [97,98]. Tax interacts with IκB kinase to promote its activity, which involves the degradation of inhibitor of nuclear factor kappa B (IκB) protein and NF-κB activation. Therefore, the persistent activation of NF-κB by HTLV-1 infection confers survival and proliferation advantages to adult human T-cell leukemia (ATL). Upon infection by HTLV-1, most naive cells experience cell-cycle arrest or senescence [99,100]. However, HTLV-1-transformed T cells and ATL display resistance to Tax protein-induced senescence [101], indicating that they have acquired genetic/epigenetic alterations that allow them to evade senescence. The hyperactivation of this oncogene has been shown to induce R-loop accumulation and DNA damage, which could contribute to genome instability in this cancer. He Y et al. demonstrated that constitutive NF-κB activation by HTLV-1 infection induces R-loop accumulation and DNA damage (Figure 3a) [56]. Nucleotide excision repair nucleases, such as XPF and XPG cleave R-loops and generate DNA double-strand breaks (DSBs), which can induce senescence. The removal of R-loops via RNase H overexpression was demonstrated to attenuate DNA damage and Tax protein-induced senescence. ATL cells are deficient in XPF, XPG, and CSB, which help to promote the proliferation of HTLV-1-infected cells by decreased DNA DSBs in ATL cells and mitigated Tax/NF-κB-induced senescence. These results implicate R-loop mediated DNA damages as critical factors for senescence resistant mechanism and ATL development.

## 7. Mutation of a Dead Box Helicase Protein Promotes Adult Myelodysplastic Syndrome (MDS)/Acute Myeloid Leukemia (AML)

DDX41 is a member of the dead box helicase protein family, which are conserved RNA-binding proteins that exhibit RNA helicase activity. Large-scale R-loop proximal proteome analysis revealed that DDX41 regulates R-loop formation [57]. Morsler et al. employed quantitative mass spectrometry (MS)-based proteomics to identify proteins that are involved in R-loop regulation. They fused ascorbate peroxidase (APEX2) to the RNA/DNA hybrid binding domain (HBD) of RNaseH1, which acts as a sensor of R-loops. After HBD binding to R-loops, APEX biotinylated R-loop-interacting proteins were further identified by mass spectrometry. Using this strategy, DDX41 was identified as an R-loop-proximal protein that functions in R-loop resolution (Figure 3b). Indeed, the depletion of DDX41 increased DNA damage and R-loop formation, whereby DDX41 is thought to directly unwind RNA/DNA hybrids through its helicase activity. Interestingly, sBLISS (Break Labeling in situ and Sequencing) was also performed to validate and identify site-specific DSBs induced by R-loop formation in DDX41-deficient cells.

In addition to the physical generation of damaged DNA, the depletion of DDX41-mediated R-loop formation and DSBs promotes an inflammatory response [57,102,103]. Weinreb J et al. observed that excessive R-loop formation triggers an inflammatory cascade, which is important for hematopoietic stem and progenitor cells (HSPC) production [103]. The hematopoietic system is maintained by multipotent HSPCs that sustain the stem cell pool via self-renewal to generate mature blood cells through multilineage differentiation. The accurate maintenance of the HSPC pool is necessary for healthy hematopoiesis throughout life. The abnormal expansion of HSPCs is a common feature in clonal hematologic malignancies including leukemia [104,105]. Inflammatory signaling plays a physiological role in regulating HSPC biology from the earliest stages of stem cell formation in the embryo and throughout the aging process [106,107]. Given that R-loop mediated inflammation promotes HSPC expansion, this finding may suggest that this pathway could affect leukemia tumorigenesis through these activities.

## 8. Mutations in Pre-mRNA Splicing Protein Promote MDS and Leukemia

Pre-mRNA splicing genes are the most common mutated genes in myelodysplastic syndromes (MDS). Over 50% of all MDS cases are associated with pre-mRNA splicing gene (*SF3B1*, *SRSF2*, *U2AF1,* and *ZRSR2*) mutations [108,109,110]. Pre-mRNA splicing converts newly generated precursor messenger RNA (pre-mRNA) transcripts into mature messenger RNA (mRNA). This process removes the intron sequence from the pre-mRNA precursor and splices the exons back together. Such gene mutations result in splicing defects, which are linked to these diseases including MDS and leukemia. Genetic studies have revealed that pre-mRNA splicing gene mutations act as a driver for cancer development. For example, SRSF2 P95H mutation knock-in mice exhibit multiple lineage dysplasia [111,112], transgenic U2AF1 S34F mutant mice develop leukopenia [113], and SF3B1 K700E knock-in mice have impaired erythroid maturation [114].

A genome-wide siRNA screen provided strong evidence that the RNA-splicing pathway is involved in the genomic instability mediated by R-loop formation [115]. The knock-down of RNA-splicing proteins was found to induce DNA damage, which was rescued by RNaseH overexpression. Cytological screening in yeast also found that RNA processing mutants induce R-loop formation [116]. The depletion of SRSF1 (ASF/SF2) induces DNA damage by R-loop accumulation (Figure 3c) [58]. In this study, DT40-ASF cell lines were used by introducing a single copy of the *ASF/SF2* gene under the control of a tetracycline repressible promoter. Upon treatment of tetracycline, the depletion of ASF/SF2 occurs, leading to cell-cycle arrest and apoptotic cell death. The overexpression of RNaseH prevented R-loop mediated cell-cycle arrest and a hypermutation phenotype in these cells. SRSF1 directly suppressed R-loop formation in an in vitro transcription assay as well. U2AF1 S34F and SF3B1 K700E mutations were found to induce DSBs through R-loop formation (Figure 3c) [59,60]. These mutant cells exhibited ATR activation by R-loop formation, which resulted in ATR inhibitor sensitivity. These results demonstrate the beneficial effect of ATR inhibitors on leukemia patients with spliceosome mutations, which may be through an R-loop-associated vulnerability. However, R-loop regulation mechanisms involved in these RNA-splicing proteins are unknown and further studies regarding the molecular mechanism of R-loop regulation, including in this disease context, are needed.

## 9. Mutation of mRNA Cleavage and Polyadenylation Proteins Associated with Eosinophilic Leukemia

A genetic study in yeast determined that mRNA cleavage and polyadenylation (mCP) proteins are involved in R-loop-mediated genome instability [61]. This study found that yeast cells with mutations in mCP machinery accumulated R-loops, which can lead to DNA damage and genome instability. The researchers focused on FIP1, the human ortholog of yeast FIP1, which is a conserved mCP component. In eosinophilic leukemia, FIP1 functions are lost because of an oncogenic fusion and knockdown of FIP1 promotes DNA damage and chromosome breakage. These results suggest that a dysfunction of mCP-mediated R-loop accumulation and genome instability may contribute to the development of eosinophilic leukemia. However, mechanistic studies will be required to identify how mCP proteins regulate R-loop formation and how these may contribute to this specific form of leukemia.

## 10. Concluding Remarks

Over the past decade, various tools have been developed to detect R-loops, which have contributed to identifying the function of both R-loops and R-loop regulators. Recent studies have demonstrated that mutated genes in cancer affect genomic stability through R-loop formation. In leukemia, DNA damage resulting from a deficiency in various DNA repair proteins contributes to cancer development [117]. In this review, we have described the effect of R-loop accumulation on genomic instability with a focus on leukemia cancers. Although it has been revealed that R-loops accumulate as a result of multiple processes, more studies are needed to understand the contribution of genetic instability to the etiology of leukemia specifically. The abnormal accumulation of R-loops results in DNA damage, but it is unclear through what mechanism R-loops and/or DNA damage production impact cancer development. Although lncRNAs, such as *HOTTIP*, do not induce DNA damage through R-loop production, they can regulate the expression of oncogenes through TAD formation. Epigenetic regulation by R-loops may affect the expression of oncogenes and tumor suppressors, apart from DNA damage; thus, it is necessary to further study R-loop production and related mechanisms in various non-coding RNAs that are altered in leukemia.

R-loops have been implicated in the response to chemotherapy in leukemia. Chemotherapeutic agents, such as topoisomerase inhibitors, can induce R-loop formation and DNA damage, leading to cell death [118,119]. These inhibitors work by trapping topoisomerase II (TOP2) enzymes on DNA, leading to the formation of TOP2 cleavage complexes which induce the accumulation of R-loops. However, it is still unclear the role of R-loops in TOP2 inhibitor-induced leukemic cell death. In our previous work, we identified that topoisomerase 2 generated DNA double-strand breaks through R-loop accumulation in cells [120]. These results implicated that TOP2 trapping on R-loop regions by inhibitors might amplify DNA damage and/or inhibit DNA repair, leading to cell death. However, these drugs have limitations because they have serious side effects, including secondary leukemia mediated by chromosome translocations. Topoisomerase inhibitor-induced DNA damage leads to chromosomal rearrangements involving the *MLL* gene on chromosome 11q23. These rearrangements typically result in the fusion of the *MLL* gene, which has been implicated in leukemogenesis [121]. Future studies are needed to identify the roles of R-loop-mediated DSBs in chromosome rearrangements. The FDA has approved the use of DNA methyltransferase inhibitors, namely azacytidine and decitabine, for the treatment of AML [122]. Shukla V et al. demonstrated that the depletion of DNMT1 can prevent the accumulation of R-loops in TET-depleted cells [55]. The regulation of R-loop homeostasis and genome stability in cancer cells by DNMT inhibitors may be a potential mechanism for the treatment of AML with azacytidine and decitabine. However, it is still unknown how DNA methylation regulates R-loop formation precisely, an answer to which might provide the mechanism of DNMT1 inhibitor cell responses for AML treatment. In conclusion, R-loops and their associated factors play important roles in the development and progression of leukemia through their involvement in gene expression regulation, as well as DNA damage responses. Further research will be required to fully understand the mechanisms by which R-loops contribute to leukemia. This information will be vital for developing targeted therapies that can modulate R-loop metabolism and their vulnerabilities in cancer cells, including but not limited to leukemias.

## Figures and Tables

**Figure 1 ijms-24-05966-f001:**
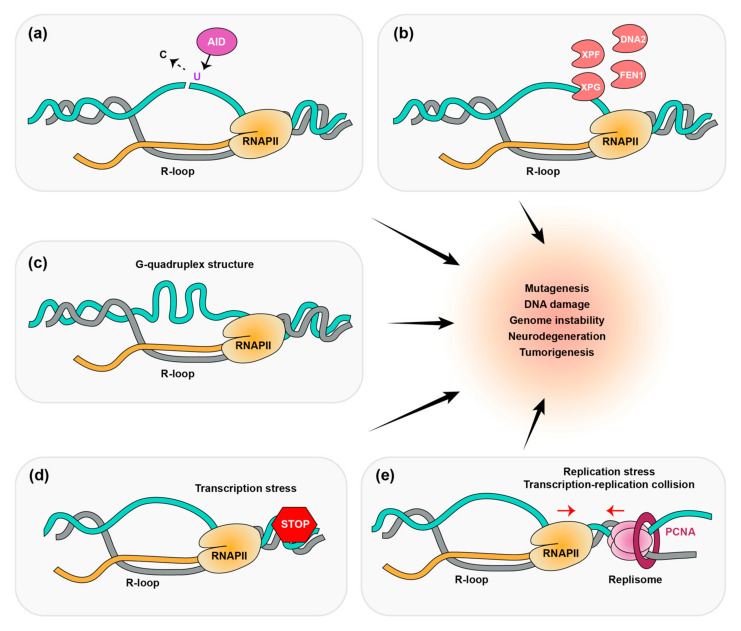
R-loops are caused by several processes that generate genomic instability. Multiple sources of R-loops induce a DNA damage response and interfere with genome integrity, which can lead to mutagenesis, neurodegeneration, and tumorigenesis. Sources of R-loops include: (**a**) displaced ssDNA on R-loops, which are nicked by AID-mediated DNA nucleases; (**b**) non-templated ssDNA, which may be cleaved by DNA nucleases including XPG, XPF, FEN1, and DNA2; (**c**) G-quadruplex secondary structures; (**d**) transcription stress; (**e**) replication stress and transcription-replication collisions (TRCs).

**Figure 2 ijms-24-05966-f002:**
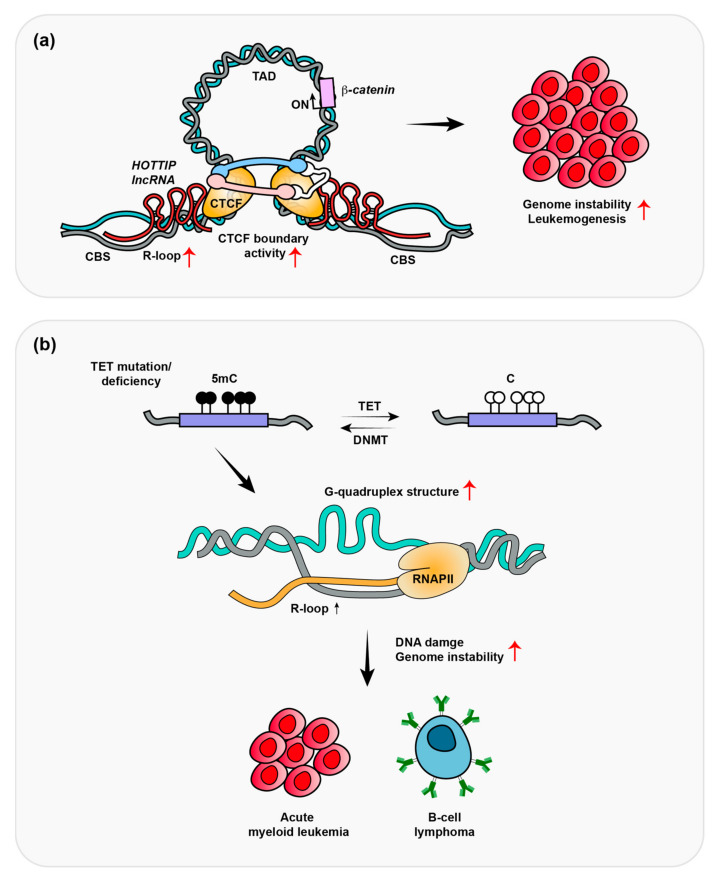
*HOTTIP* lncRNA and TET enzymes regulate R-loop formation. (**a**) Elevated *HOTTIP* lncRNA results in R-loop formation and promotes CTCF boundary activity and TAD formation to drive oncogene transcription, which are linked to the induction of genomic instability and leukemogenesis. (**b**) The loss of TET function blocks the conversion of 5mC to 5hmC and increases G-quadruplex structures. TET deficiency promotes R-loop accumulation and tumorigenesis, including in AML and B-cell lymphoma.

**Figure 3 ijms-24-05966-f003:**
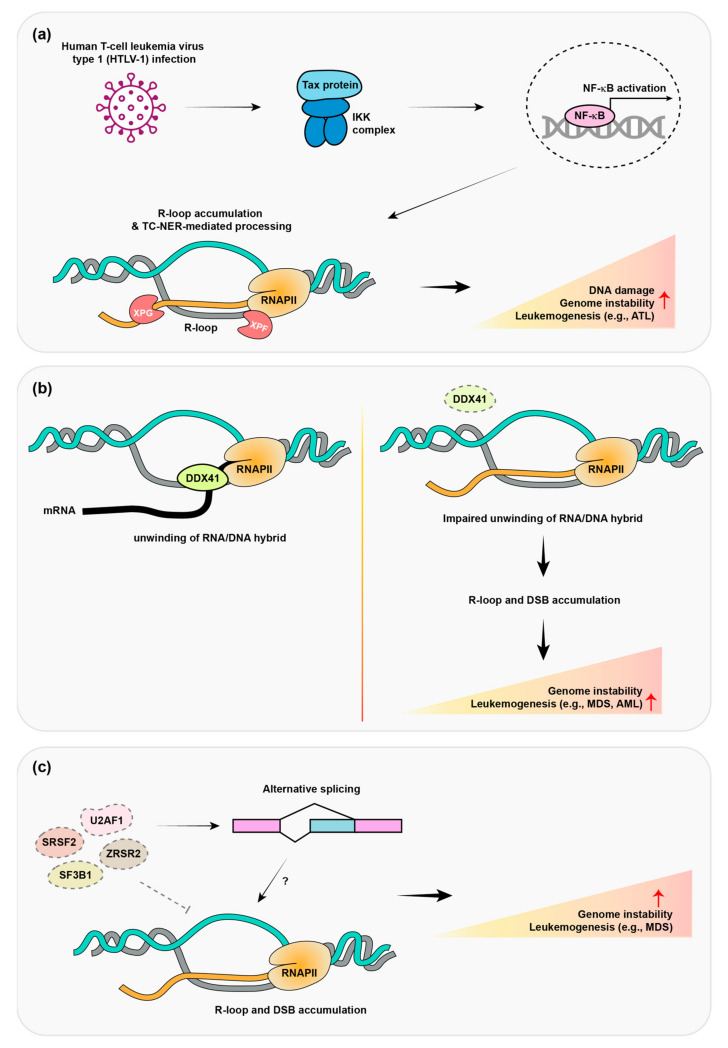
HTLV-1 infection, DDX41, and pre-mRNA splicing factors mediate R-loop formation. (**a**) Following HTLV-1 infection, the viral Tax protein activates the NF-kB signaling pathway. The hyperactivated oncogene promotes R-loop formation and contributes to genomic instability, which can give rise to the leukemogenic process; (**b**) Mutation of DDX41 impairs unwinding of the RNA/DNA hybrid (R-loop) using its helicase activity and generates a DSB-mediated inflammatory response. This may enhance the insufficient self-renewal of HSC and progenitor cells (HSPCs) to trigger leukemogenesis; (**c**) The mutation of pre-mRNA splicing factors results in alternative splicing and an increase in R-loop formation and DNA strand breaks, which may lead to the development of leukemia.

**Table 1 ijms-24-05966-t001:** R-loop regulators and related mechanisms in leukemia.

**Types**	**R-Loop Regulators**	**Status**	**Description**	**Reference**
lncRNA	*HOTTIP*	Upregulated in MDS	Forms an R-loop structure at CBS sites for TAD formation and oncogene expression	[54]
Protein	TET2, TET3	Mutation	Increases DNA methylation to promote R-loop formation	[55]
Virus	HTLV-1	Infection	Constitutively activates NF-kB to promote R-loop formation	[56]
Protein	DDX41	Mutation	Unwinds RNA–DNA hybrids	[57]
Protein	SRSF1	Mutation	Splicing factor that binds to Pol II CTD and suppresses R-loop formation	[58]
Protein	U2AF1	Mutation	Splicing factor that suppresses R-loop formation, but the mechanism is unknown	[59]
Protein	SF3B1	Mutation	Splicing factor that suppresses R-loop formation, but the mechanism is unknown	[60]
Protein	FIP1L1	Mutation	mRNA cleavage and polyadenylation(mCP) protein suppresses R-loop formation	[61]

## Data Availability

Not applicable.

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
