# Peer review of "Clinical and Mechanistic Implications of R-Loops in Human Leukemias"

_ijms, 2023, doi:10.3390/ijms24065966_

Round 1

Reviewer 1 Report

The review article describes recent studies that reveal the role of R-loops in human leukemia and the functional relevance between R-loops and cancer initiation. Summarizing the findings so far may provide a better understanding of the disease and provide new strategies for treating of leukemia. The topic of the article is appropriate for the journal and the Special issue.

Some aspects of the manuscript need to be improved before the final acceptance of the paper

1. Table 1 must be formatted according to requirements.

2. Figures are not cited in the main text.

3. The section “Author contributions” must be included

4. The page range is not written properly in some of the references.

5. It is necessary to remove the grammatical errors that are present - lines 94, 119, 124, 145, 294, etc.

Reviewer 2 Report

Authors have focused on Clinical and Mechanistic implications of R-loops in Human LeukemiaThe concept of MS is good. However, the manuscript can be improved by adding Materials and Methods section.

The specific comments, which could help to improve the manuscript, are:

1.      Highlight text by yellow colour related to Clinical implications given in manuscript. Just for review purposes.

2.      Methodology is missing in Abstract.

3.      Add clear inclusion and exclusion criteria for selection of articles under methodology section. Why are articles seem relevant not included in review? Like, Transcription-coupled genetic instability marks acute lymphoblastic leukemia structural variation hotspots. Elife. 2016 Jul 19;5:e13087. doi: 10.7554/eLife.13087.

4.     Authors are encouraged to add information related to effect of FDA approved drugs (if available in literature) on R-loops associated concept on leukaemia.

5.      A significant discussion part is missing in Manuscript.

6.      Page 3, line 117-118: give examples of cancers here????

7.      Why have authors not included eosinophilic leukemia (FIP1)? Refer to the suggested references.

Richard P, Manley JL. R Loops and Links to Human Disease. J Mol Biol. 2017 Oct 27;429(21):3168-3180. doi: 10.1016/j.jmb.2016.08.031. 

Stirling PC, Chan YA, Minaker SW, Aristizabal MJ, Barrett I, Sipahimalani P, Kobor MS, Hieter P. R-loop-mediated genome instability in mRNA cleavage and polyadenylation mutants. Genes Dev. 2012 Jan 15;26(2):163-75. doi: 10.1101/gad.179721.111.

8.      It would be better to add research gap and future prospects related to the topic in conclusion section.
